# Ultraviolet Radiation Sensor Based on ZnO Nanorods/La_3_Ga_5_SiO_14_ Microbalance

**DOI:** 10.3390/s21124170

**Published:** 2021-06-17

**Authors:** Dmitry Roshchupkin, Arkady Redkin, Eugenii Emelin, Sergey Sakharov

**Affiliations:** 1Institute of Microelectronics Technology and High Purity Materials, Russian Academy of Sciences, Academician Ossipyan St. 6, 142432 Chernogolovka, Russia; arcadii@iptm.ru (A.R.); eemelin@iptm.ru (E.E.); 2FOMOS Materials Co., Buzheninova St. 16, 105023 Moscow, Russia; sakharov@newpiezo.com

**Keywords:** ZnO nanorods, La_3_Ga_5_SiO_14_ crystal, microbalance, UV sensor, X-ray diffraction, bulk acoustic wave, scanning electron microscopy

## Abstract

The possibility of creating resonant ultraviolet (UV) sensors based on the structure of ZnO nanorods/La_3_Ga_5_SiO_14_ microbalance (LCM) has been investigated. The principle of sensor operation is based on the desorption of oxygen from the surface of ZnO nanorods upon irradiation with UV light and an increase in the concentration of charge carriers that leads to an increase in the capacitance of the structure of ZnO nanorods/LCM. It has been shown that UV radiation intensity affects the resonance oscillation frequency of the LCM sensor. After the end of irradiation, the reverse process of oxygen adsorption on the surface of ZnO nanorods occurs, and the resonance frequency of the sensor oscillations returns to the initial value.

## 1. Introduction

Zinc oxide (ZnO) is a wide bandgap semiconductor (bandgap width 3.37 eV) with good piezoelectric properties. The crystal structure of ZnO has a point symmetry group of 622. There are a large number of methods for obtaining zinc oxide on various substrates in the form of films or ordered layers of nanocrystals in the literature. Often, the method of hydrothermal synthesis from solution is used for this purpose [1,2,3], which allows layers of monocrystalline nanorods (1D crystals) to grow at a relatively low temperature. To obtain textured ZnO films, the method of magnetron sputtering of Zn in an oxygen atmosphere has proven itself well [4,5,6]. Such films can be used in acoustoelectronics as a piezoelectric material for excitation and propagation of surface or bulk acoustic waves [7]. The polar axis c is usually normal to the substrate surface in the obtained arrays of nanorods and textured ZnO films. Ordered arrays of ZnO nanorods obtained by gas phase synthesis methods (CVD) are of great interest [8,9,10]. One-dimensional crystals obtained by this method have a perfect structure that allows them to be used as optoelectronic devices [11], piezogenerators [12], solar cells [13], etc. Recently, there has been great interest in the creation of various sensors. One of the promising applications of zinc oxide is a creation of gas sensors using arrays of nanocrystals [14,15,16,17,18,19], as well as single ZnO nanorods [20,21].

The electrophysical characteristics of ZnO are known to be sensitive to UV radiation. According to the generally accepted concepts, the mechanism of the effect of UV radiation on the electrical conductivity of zinc oxide is as follows. Under normal conditions, the ZnO surface is covered with adsorbed oxygen, the molecules of which capture electrons from the surface layer, converting into $O_{2}^{-}$, $O^{-}$ and $O^{2-}$ ions. As a result of this process, the concentration of charge carriers in the bulk of the materials decreases. In addition, the formation of a layer of negatively charged ions on the surface leads to the displacement of electrons and the formation of a charge-depleted zone in the near-surface layer. Electron–hole pairs arise in zinc oxide upon irradiation with UV light. Oxygen ions interact with holes to form neutral $O_{2}$ molecules, which are desorbed from the surface. In this case, the concentration of electrons increases, and the depth of the depleted zone decreases, which leads to a decrease in the electrical resistance [22]. It should be noted that this process is reversible. In the dark, the opposite process takes place: oxygen adsorption on the ZnO surface and restoration of the original electrical properties. This phenomenon is used to create resistive sensors that allow UV radiation to be recorded by changing electrical resistance [23,24].

At the same time, the change in the charge state in ZnO upon UV irradiation can be detected in other ways. In [25], surface acoustic wave (SAW) devices were used for these purposes. One of the options for creating UV radiation sensors is the use of a combination of ZnO and resonant sensors based on bulk acoustic waves (QCM, quartz crystal microbalance; LCM, La_3_Ga_5_SiO_14_ langasite crystal microbalance). In [26], such a UV radiation sensor was implemented, in which the LCM was used. A ZnO film was formed on the LCM surface by magnetron sputtering. Under the influence of UV radiation, oxygen is desorbed from the film surface, the result of which is the appearance of additional charges in the ZnO film, which change the resonance excitation frequency of bulk acoustic waves in the LCM. The principle of operation of the resonant sensor is based on the fact that the LCM is schematically an L–C–R electrical circuit. The ZnO film acts as a capacitor that is parallel to the resonator’s own capacitor. The capacities of two parallel capacitors are added. An increase in the capacitance of the ZnO film under the influence of ultraviolet radiation leads to an increase in the total capacitance of ZnO/LCM and a corresponding decrease in the resonance excitation frequency of the LCM. The change in resonance frequency of LCM depends on the exposure intensity of UV radiation. Note that in the future, the use of the LCM will allow wireless UV sensors to be created.

In this work, the possibility of forming ZnO nanorods on the LCM surface has been investigated. The application of ZnO nanorods allows the use of a significantly larger active surface in LCM UV sensors in comparison with the ZnO film formed by magnetron sputtering that can increase the sensitivity of the UV sensor.

## 2. Fabrication of ZnO Nanorods/LCM Structure

Figure 1 shows a photograph from both sides of an LCM that was used as a resonator to create a UV sensor. For the fabrication of the resonator, the yxl/+45°-cut of an LGS crystal was used, which is associated with the subsequent formation of a ZnO film or an array of ZnO nanorods on the resonator surface. The use of piezoelectric quartz SiO_2_ as a QCM is not possible, since during the formation of ZnO films or nanorods, it is necessary to use a high temperature of ~600 °C (annealing of the ZnO film, CVD synthesis of nanorods), which exceeds the phase transition temperature in quartz. The transition of α-quartz to β-quartz occurs at a temperature of 573 °C. This process is irreversible, so using quarts to fabricate UV sensors, in this case, is difficult. Using crystals of the langasite family is optimum to fabricate resonators (microbalances). The phase transition temperature in these crystals is ~1450 °C. A number of crystal cuts, including yxl/+45°-cut, have almost a zero temperature coefficient of frequency. The standard resonator diameter is 14 mm.

Al and Ag are usually used to fabricate electrodes on resonator surfaces. However, these materials are not suitable for high-temperature processes to obtain ZnO films and nanorods on resonators surfaces due to using a high temperature and oxygen environment that leads to the oxidation of electrodes and degradation of their electrical properties. In this work, platinum was used to fabricate electrodes on the LCM surface. Pt films with a thickness of 100 nm were formed by magnetron sputtering on the resonator surfaces. Pt electrodes are stable during the synthesis of films and ZnO nanorods that allows them to be used to create UV sensors.

Ordered layers of ZnO nanorods were deposited using the previously developed gas-phase method, in which the growth of 1D crystals was carried out due to the “self-catalytic” VLS (vapor–liquid–crystal) process [8]. The synthesis was carried out in a two-zone quartz flow-type reactor at reduced pressure. Zinc metal was placed in the first zone, and substrates (LCM) in the second zone. The operating temperature of the first zone (T_1_) was 620 °C, while the temperature of the second zone (T_2_) was 570 °C. Zinc was evaporated in the first zone. Zinc vapors fell into the second, less heated zone, where they partially condensed on the surface of the substrates. An oxygen-argon mixture (10% O_2_) also entered this zone. As a result of the reaction of zinc with oxygen, zinc oxide nanorods grew. The growth diagram of ZnO nanorods is shown in Figure 2.

At the initial stage, the temperature in the evaporation and growth zones was brought to operating values. In this case, oxygen was not supplied to the growth zone. Therefore, due to the temperature difference between the evaporation zone and the growth zone, an array of liquid zinc microdroplets with a diameter of ~100 nm was formed on the surface of the Pt electrode (stage 1). In stage 2, an oxygen-argon mixture was injected into the growth zone. As a result of the reaction with oxygen, an oxide was formed in the zinc microdrop, which diffused to the substrate surface and crystallized under the Zn microdrop (at the Zn microdrop/Pt electrode interface). Further, the growth of ZnO nanorods occurred by the VLS mechanism along the normal to the surface of the Pt electrode. At the third stage, the growth of ZnO nanorods continued in the stationary mode due to the inflow of Zn vapors and oxygen into the growth zone. At the final stage (4), the temperature in the evaporation zone was reduced and the oxygen supply was stopped, as a result of which Zn evaporated from the tops of the nanorods. Figure 3 shows microphotographs of ZnO nanorods grown by VLS on the surface of Pt electrode. With a diameter of ~100 nm, the nanorods are approximately 10 μm long. As can be seen from the micrographs, the nanorods have a hexagonal facet that corresponds to growth along the polar axis c and point symmetry group 622.

To study the structure of ZnO nanorods, the high-resolution X-ray diffraction spectrum of ZnO nanorods was measured on a four-circle Bruker D8 Discover diffractometer with a rotating copper anode (Cu$K_{\alpha 1}$ radiation, λ=1.54 Å) in a doble–crystal X-ray diffractometer scheme. Figure 4 schematically shows a double-crystal X-ray diffractometer, used to measure the diffraction spectra of ZnO nanorods. X-ray radiation was collimated with an entrance slit of 1 mm. Then, X-ray radiation was monochromatized using two double Ge(220) monochromators in the Barter scheme. After the monochromators, X-ray radiation was re-collimated with a 1 mm slit. Then, X-ray radiation falls on the surface of the Pt electrode of LCM covered by ZnO nanorods at an incidence angle θ. The diffracted X-ray radiation was recorded with a standard NaI scintillation detector with a 1 mm entrance slit. The diffraction spectrum was measured in an angular scanning scheme 2θ−θ, where the detector is rotated by an angle of 2Θ when the sample is rotated by an angle of θ.

Figure 5 shows an X-ray diffraction spectrum of ZnO nanorods, on which (002) and (004) reflections can be observed. This means that in our case, the ZnO nanorods are normal to the surface of the LCM and oriented along polar axis c.

## 3. Results and Discussion

The use of the ZnO nanorods/LCM UV sensor is based on the change in the resonance excitation frequency of the LCM under the influence of UV radiation on the ZnO nanorods. Figure 6 shows the amplitude-frequency response of ZnO nanorods/LCM, on which the resonance can be observed at a frequency of f=5.893 MHz and antiresonance—at a frequency of f=5.902 MHz. The insets in the figure show photomicrographs of the acoustic wave field distribution in LCM resonators in real time mode, obtained by scanning electron microscopy (SEM) at resonance and antiresonance. To visualize the acoustic wave field in LCM, the SEM method in the mode of the low energy secondary electron registration was used, because secondary electrons with energy of ~1 eV are sensitive to the electric field that accompanies the propagation of acoustic waves in piezoelectric crystals [27,28,29]. It is clearly seen that the distribution of the acoustic wave field under resonance and antiresonance conditions is very different in the LCM resonator.

Figure 7 shows a scheme to investigate the properties of a UV radiation sensor based on a ZnO nanorods/LCM system. The principle of the sensor is based on the fact that the equivalent circuit of the LCM resonator is an L–C–R electrical circuit (L is inductance, C is capacitance, R is resistance). The presence of nanorods on the surface of the Pt electrode can be seen as an additional capacitor embedded in parallel to the capacitor of the LCM resonator. In this case, when two capacitors are connected in parallel, their capacitances are added up. The formation of additional charges under the influence of UV radiation leads to an increase in the capacitance of a capacitor based on ZnO nanorods. As shown in Figure 7, ZnO nanorods on the surface of the Pt electrode were irradiated with a UV lamp with a wavelength of 240 ÷ 360 nm and a power density of 300 mW/cm^2^. The distance l from the UV lamp to the LCM varied from 15 to 30 cm. With increasing l, the irradiation intensity decreases and, respectively, the amount of charges formed in the ZnO nanorods decreases and the capacitance of the capacitor decreases. The amplitude-frequency response of the ZnO nanorods/LCM was recorded using a spectrum analyzer.

Figure 8 shows the amplitude-frequency responses of ZnO nanorods/LCM measured at different distances l from the UV radiation source. Increasing the distance l leads to the radiation intensity decreasing. It can be seen from the figure that a change in the radiation intensity leads to a change in the resonance excitation frequency of the LCM. Reducing the distance between the UV sensor and the UV radiation source leads to an increase in the irradiation intensity of the ZnO nanorods. This, in turn, leads to an increase in the desorption of oxygen from the surface and an increase in the concentration of charges in the nanorods. An increase in the charge concentration leads to an increase in the capacitance of the ZnO nanorod layer on the LCM surface. The total capacitance of the two parallel capacitances increases (the capacitance of the ZnO nanorods and the capacitance of the LCM), which eventually leads to a decrease in the resonance excitation frequency of the LCM resonator. Thus, as the distance between the UV radiation source and the UV sensor decreases, the resonance excitation frequency of the LCM resonator decreases. After switching off the UV radiation source, the amplitude-frequency response of the LCM returns to its initial state due to charge compensation during the adsorption of oxygen on the surface of the nanorods. The return time to its initial state is ~30 s.

Figure 9 demonstrates the dependence of the resonance excitation frequency of ZnO nanorods/LCM sensor on the distance l to the UV radiation source. It is observed that the change in frequency from the distance l to the UV radiation source has a nonlinear character, which is determined by the absorption of UV radiation in the atmosphere. As the distance increases, the change of the sensor frequency decreases, because the absorption of UV radiation increases.

Thus, the studied ZnO nanorods/LCM system is a good option for a UV radiation sensor. The use of LCM in the future will make it possible to create wireless UV sensors.

## 4. Conclusions

A resonant UV sensor based on ZnO nanorods/LCM has been investigated. The principle of the sensor operation is based on the desorption of oxygen from the surface of ZnO nanorods and an increase in the concentration of charges, which leads to an increase in the capacitance of the ZnO nanorods/LCM system and a corresponding decrease in the resonance excitation frequency of the LCM resonator. This process is reversible, since in the absence of UV radiation, oxygen is absorbed on the surface of ZnO nanorods and the system returns to the initial charge state.

Note that the application of the LCM is associated with an almost zero temperature coefficient of frequency. Using ZnO nanorods instead of films allows the sensitivity of the UV radiation sensor to be increased by increasing the surface, on which oxygen desorption occurs under the influence of UV radiation.

Moreover, it should be noted that the application of LCM makes it possible to use sensors for operation at high temperatures. Since the piezoelectric moduli of langasite crystals exceed the corresponding values in quartz crystals, it is possible to implement wireless UV radiation sensors in the future.

## Figures and Tables

**Figure 1 sensors-21-04170-f001:**
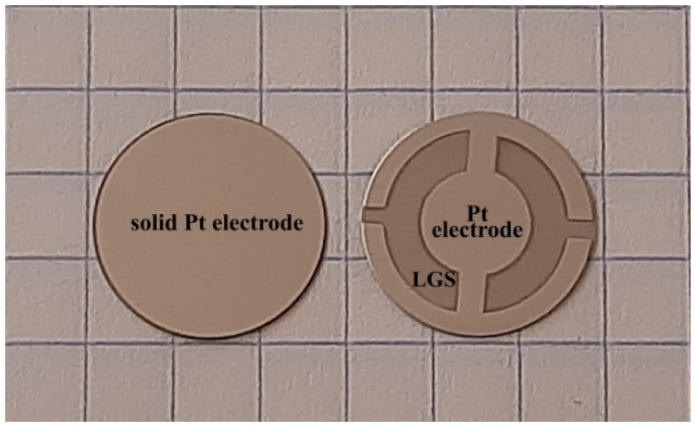
LCM based on the yxl/+45°-cut of an LGS crystal (Pt electrodes).

**Figure 2 sensors-21-04170-f002:**
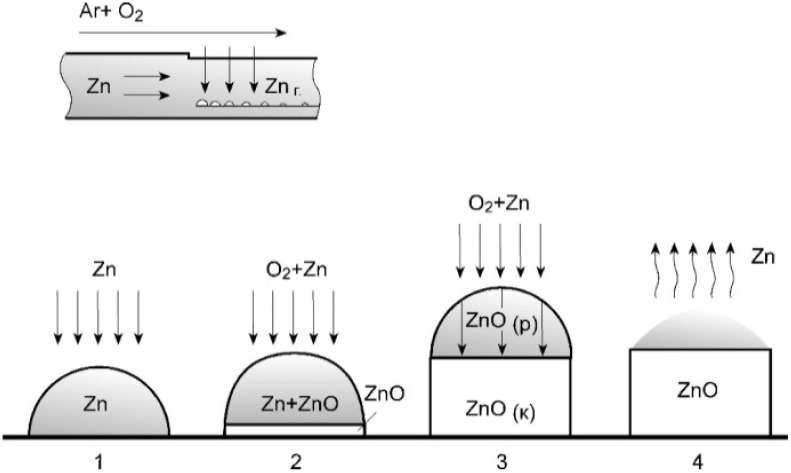
VLS growth of ZnO nanorods: 1—formation of zinc droplets (beginning of the process), 2—nucleation of the ZnO crystal, 3—growth of the ZnO nanorod, 4—evaporation of the zinc drop (end of the process).

**Figure 3 sensors-21-04170-f003:**
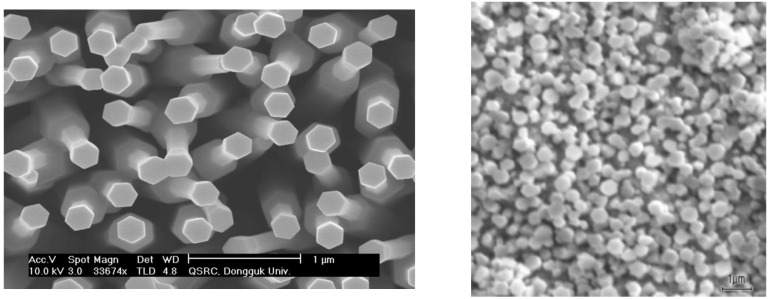
ZnO nanorods grown on the surface of Pt electrode.

**Figure 4 sensors-21-04170-f004:**
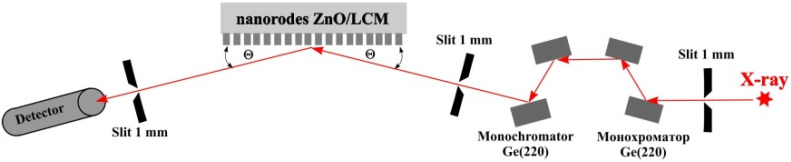
Experimental set-up of a double-crystal X-ray diffractometer.

**Figure 5 sensors-21-04170-f005:**
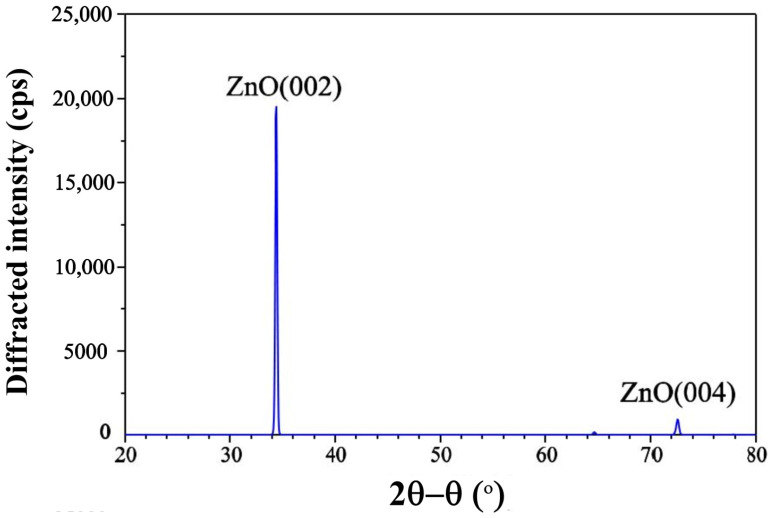
XRD spectrum of ZnO nanorods grown by VLS method on the surface of Pt electrode of LCM.

**Figure 6 sensors-21-04170-f006:**
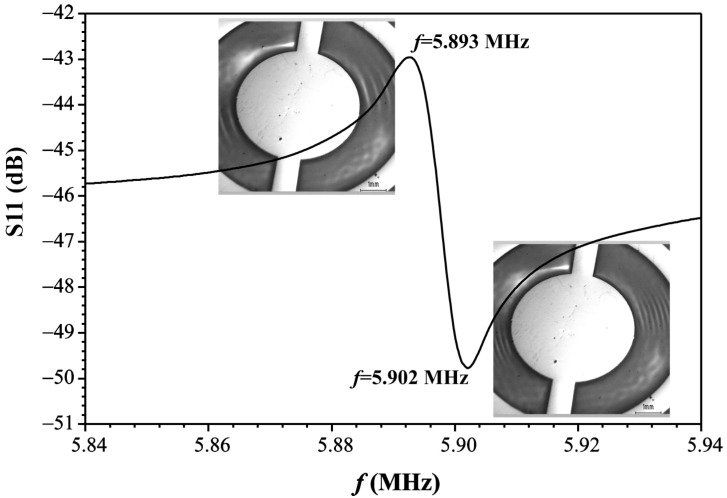
Amplitude-frequency response (S11) of the ZnO nanorods/LCM (resonance–antiresonance).

**Figure 7 sensors-21-04170-f007:**
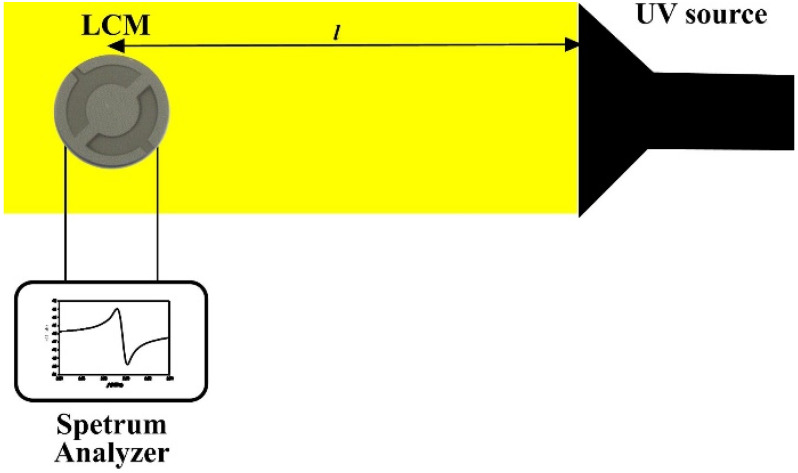
Experimental scheme of the investigation of the ZnO nanorods/LCM UV sensor characteristics.

**Figure 8 sensors-21-04170-f008:**
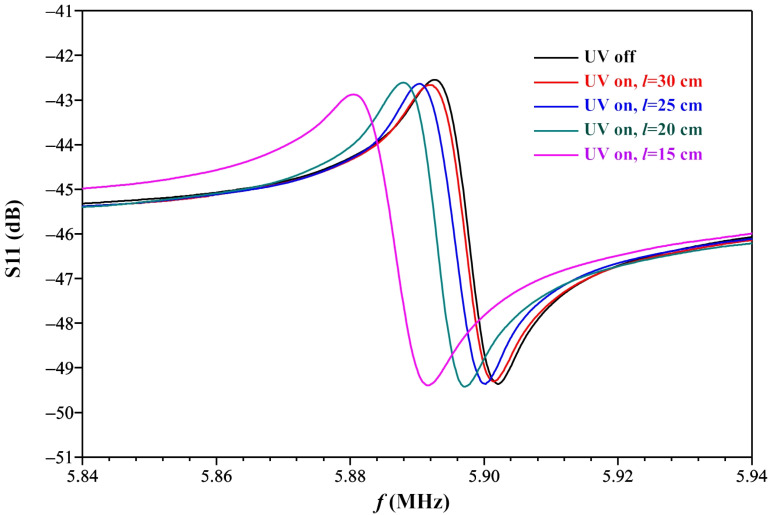
Variation of S11 characteristics of ZnO nanorods/LCM sensor under the influence of UV radiation versus distance l from the UV radiation source to the UV sensor.

**Figure 9 sensors-21-04170-f009:**
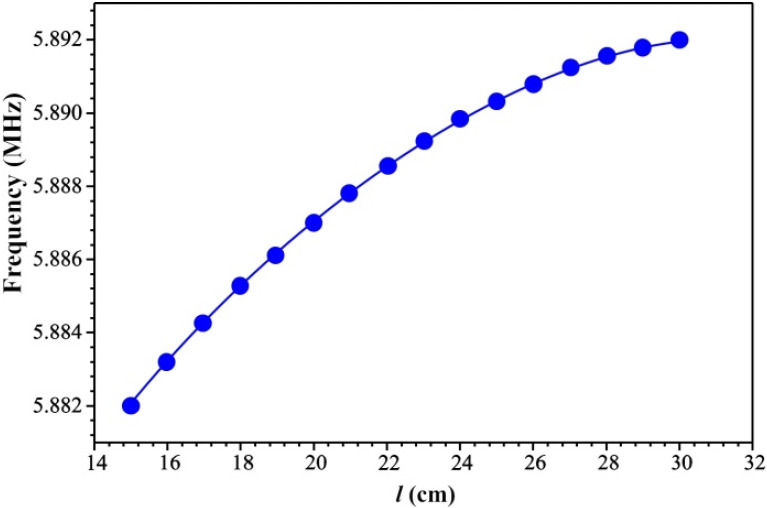
Dependence of UV sensor frequency versus distance l from the UV radiation source.

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
