# Peer review of "Ultraviolet Radiation Sensor Based on ZnO Nanorods/La3Ga5SiO14 Microbalance"

_sensors, 2021, doi:10.3390/s21124170_

Round 1
Reviewer 1 Report
The authors reported an ultraviolet radiation sensor. The paper was well written; the experimental resultsare interesting; the analysis and discussion of the experimental results are rational and helpful to readers.
However, I think that the manuscript has few results to consider a full article. I suggest being a short communication.
Before its acceptance, some issues should be addressed:
1) p.1 - line 43
There is a symbol и.
2) Figure 1
Authors need to identify platinum and crystal in the image.
3) Results of Figure 8.
Did the authors graph the frequency ratio as a function of distance? Is there a linear relationship?
4) As a curiosity. Did the authors study the influence of light intensity on the sensor signal? As the distance influences, logically the intensity will influence. Was this study carried out?
Author Response
Response to Reviewer 1 Comments
Dear reviewer, thank you for your correct and helpful questions. In the future, we plan to continue research in this direction.
Point 1: p.1 - line 43
There is a symbol и.
Response 1: Symbol “и” was replaced by “and”.
Point 2: Figure 1
Authors need to identify platinum and crystal in the image.
Response 2: The platinum and LGS crystal are identified in the figure 1.
Point 3: Results of Figure 8.
Did the authors graph the frequency ratio as a function of distance? Is there a linear relationship?
Response 3: Additionally we inserted Figure 9, which shows the dependence of the frequency of UV sensor versus distance from UV source. This dependence has a nonlinear character and is associated with the absorption of ultraviolet radiation in air. As the distance increases, the change in the frequency of the sensor oscillations decreases.
Point 4: As a curiosity. Did the authors study the influence of light intensity on the sensor signal? As the distance influences, logically the intensity will influence. Was this study carried out?
Response 4: At present, we used a low-power UV radiation source and did not conduct research on the influence of radiation intensity on the frequency characteristics of the sensor. Definitely there will be a dependence of the frequency of sensor oscillation versus UV intensity. Increasing of UV intensity leads to decreasing of the frequency due to increasing of the oxygen desorption from the surface of ZnO nanorods.
In fact, we plan to continue research using synchrotron radiation source, where it will be possible to use both different distances from the source to the sensor and to studied UV sensor properties at different UV wavelengths. Moreover, there is also interest in studies using soft X-rays.
Unfortunately, the possibility of research at the synchrotron light source BESSY II has been postponed since February 2020. The experiment was supposed to take place at the end of March 2020 (after in April 2021), but it has been pushed back today to somewhere around 2022 due to COVID-19.

Reviewer 2 Report
The paper reports a study of a particular radiation sensors for UV based on ZnO.
General comment:
The paper is well written and easily readable. The paper is also well balanced among introduction, description of the fabrication process, results and conclusion.
However, I have general comments that should deserve an answer by the authors:
- 1) The papers does not cover the aspects of the radiation damage. I guess that the UV exposition, depending on the intensity, will also degrade the sensors. I would expects at least some comments related to the foreseen radiation damage.
- 2) I am wondering if the sensors might be sensitive to other radiation spectra. In other words, if the sensors work for UV they can be ignited also by soft X-rays, for example. If not, please specify how and why you choose only UV applications and how you would discriminate sensor stimuli originated from UVs or other spectra. Please comment.
- 3) If the sensors require the air for annealing process, what will happen in vacuum (space) conditions? Any idea?
- 4) When you mention wireless applications be careful because you will need electronics to create a transmission protocol and antennas to irradiate the signal. These consume power and this aspect can dominate the entire sensor system. Please comment.
Please consider the points above and evaluate if it makes sense to address them into the paper or just reply to the reviewer.
Specific comments
- line 43: "and" symbol instead of "и" ?
I recommend a minor revision the manuscript by addressing the comments before accepting it for the publication
Author Response
Response to Reviewer 2 Comments
Dear reviewer, thank you for your correct and helpful comments.
Specific comments
- line 43: "and" symbol instead of "и" ?
Symbol “и” was replaced by “and”
Point 1: The paper does not cover the aspects of the radiation damage. I guess that the UV exposition, depending on the intensity, will also degrade the sensors. I would expects at least some comments related to the foreseen radiation damage.
Response 1: Unfortunately, we have a low-intensity UV radiation source. At the moment it is difficult to assess the level of radiation damage. But it should be noted that langasite crystal has the good temperature properties. Previously, we studied the structural of langasite crystals at synchrotron radiation source. The use of a white beam for X-ray topography did not lead to any serious changes in the crystal structure and damage. We also investigated the X-ray diffraction process on a silicon crystal covered ZnO film for surface acoustic waves propagated. The studies were carried out using both hard X-rays with an energy of 10 keV and soft X-rays with an energy of 100 eV. Synchrotron radiation had no significant effect on the structure of the ZnO film. A serious effect was produced by the propagation of the surface acoustic wave, which built up the texture of the film, i.e., there was an improvement in the piezoelectric moduli of the film.
Point 2: I am wondering if the sensors might be sensitive to other radiation spectra. In other words, if the sensors work for UV they can be ignited also by soft X-rays, for example. If not, please specify how and why you choose only UV applications and how you would discriminate sensor stimuli originated from UVs or other spectra. Please comment.
Response 2: The sensors will also be sensitive to soft X-rays. In fact, we plan to make experiments at the BESSY II synchrotron radiation source, where we can realize research in the area of UV radiation and soft X-rays. And it is interesting to carry out investigations We planned these studies at synchrotron radiation back in April 2020, but because of the pandemic everything is postponed and now for users the time has been moved to 2022.
We have not detected any effect of infrared radiation.
Point 3: If the sensors require the air for annealing process, what will happen in vacuum (space) conditions? Any idea?
Response 3: Under vacuum conditions, the oxygen desorption process will definitely have a limit. The sensor will work until all the oxygen leaves the surface of ZnO nanorods. This, too, is interesting to investigate at a synchrotron radiation source, where studies using soft X-rays and ultraviolet radiation are performed in a vacuum. It is a shame that not everything can be investigated yet because of current limitations.
Point 4: When you mention wireless applications be careful because you will need electronics to create a transmission protocol and antennas to irradiate the signal. These consume power and this aspect can dominate the entire sensor system. Please comment.
Response 4: Piezomodules in langasite crystals are an order of magnitude higher than in quartz crystals. Langasite crystal is now a good material for wireless sensors. Several years ago we participated in the 7th framework program on creation of high-temperature sensors based on LCM (langasite microbalance) for temperature range 400-600 ºÐ¡. Antenna was fabricated on the surface of LCM for receiving-transmitting the signal. On a quartz crystal this will not work because of the small piezomodules. We did some research and determined reliable values of piezomoduli in crystals of langasite family (D. Irzhak and D. Roshchupkin, Measurement of independent piezoelectric moduli of Ca3NbGa3Si2O14, La3Ga5.5Ta0.5O14 and La3Ga5SiO14 single crystals, J. Appl. Cryst. (2018). 51, 1174-1181, https://doi.org/10.1107/S1600576718009184). And the electronics will have to be made.
It is also interesting to use ZnO nanorods as a piezogenerator. In the long run it will probably be possible to make some perspective devices, consisting of a sensor and a piezogenerator.
